# Measuring Serotonin Binding to Its Receptors In Vitro via Charge Transfer to ANAP

**DOI:** 10.3390/ijms262210815

**Published:** 2025-11-07

**Authors:** Olivia G. Brado, Aspen T. Hawkins, Adam D. Hill, Michael C. Puljung

**Affiliations:** 1Department of Chemistry, Trinity College, Hartford, CT 06106, USA; obrado@berkeley.edu (O.G.B.); athawk@email.unc.edu (A.T.H.); adam.hill@trincoll.edu (A.D.H.); 2Neuroscience Program, Trinity College, Hartford, CT 06106, USA

**Keywords:** serotonin, dopamine, fluorescence, ion channel, receptor

## Abstract

Serotonin (5-HT) is a vital intercellular messenger with diverse signaling functions throughout the human body. We have characterized and implemented a novel, in vitro fluorescence-based method of measuring 5-HT binding to gain a fuller understanding of the interactions between 5-HT and its receptors. This method involves expression of 5-HT receptor proteins in cultured cells with the fluorescent, non-canonical amino acid l-3-(6-acetylnaphthalen-2-ylamino)-2-aminopropanoic acid (ANAP) incorporated into the ligand binding site. ANAP fluorescence was quenched in solution by both 5-HT and dopamine. Time-resolved photoluminescence and transient absorption spectroscopy confirmed that ANAP quenching by 5-HT occurs via a charge-transfer process that recovers through back-electron transfer on the nanosecond timescale. Supported by density functional theory calculations, this process likely involved an ANAP reduction by 5-HT. To test this method on intact receptors in a cellular context, we expressed 5-HT_3A_ receptors (5-HT-gated ion channels) in HEK293T cells with ANAP inserted co-translationally into the transmitter binding site. Fluorescently labeled 5-HT_3A_ receptors were functional and activated by 5-HT, as assessed by whole-cell patch clamp. Addition of 5-HT caused a concentration-dependent quenching of fluorescence from ANAP-tagged channels in intact cells and unroofed plasma membranes, demonstrating the utility of this method for measuring 5-HT binding to its receptors. Collectively, these results delineate a technique for measuring transmitter binding that can be widely adopted to explore 5-HT binding not only to 5-HT_3_ receptors, but to any 5-HT receptor, transporter, or binding protein in heterologous expression systems.

## 1. Introduction

Serotonin (5-hydroxytryptamine, 5-HT) is a near-ubiquitous intercellular messenger, playing crucial roles in blood, the gut, and the central nervous system (CNS) [1]. In the CNS, 5-HT is primarily produced in the raphe nuclei of the brainstem, which project widely throughout the brain and spinal cord. Through these projections, 5-HT modulates many diverse neurobiological phenomena including arousal, mood, and pain transmission [2,3]. Serotonergic dysfunction has been linked to numerous disease states including major depressive disorder, drug use disorder, and inflammatory bowel disease. To fully understand the function of 5-HT receptors and their dysfunction in these disease states, it is crucial to better understand the process of ligand binding and the energetic coupling between binding and receptor activation. Existing methods for measuring 5-HT binding to its receptors fall short of providing direct, real-time information about receptor occupancy. In this work, we propose, fully characterize, and employ a novel fluorescence-based method that directly reports the binding of 5-HT to fluorescently tagged receptors in real time.

5-HT binds to a diverse group of receptor molecules. In humans, 5-HT receptors are grouped into seven families (5-HT_1_ through 5-HT_7_), based on sequence homology [1]. Each family encompasses several receptor subtypes. With the exception of the 5-HT_3_ family of receptors, 5-HT receptors are G-protein coupled; binding of 5-HT prompts dissociation of heterotrimeric G-proteins from its receptors, triggering downstream, intracellular signaling events. In contrast, 5-HT_3_ receptors (5-HT_3A_-5-HT_3E_ in humans) are part of the pentameric ligand-gated ion channel (PLGIC) superfamily [4,5]. Binding of 5-HT opens an aqueous pathway for cation flow across the plasma membrane, generating an excitatory current. 5-HT_3_ receptors are important regulators of brain–gut signaling and the vomiting response and have been implicated in several diseases including inflammatory bowel disease (IBS), schizophrenia, and substance use disorder [6,7,8].

To gain a richer understanding of structure–activity relationships in 5-HT receptors, we sought to develop an assay that allows for real-time measurement of transmitter binding to its receptors in heterologous expression systems. Equilibrium binding assays operate over long time scales and non-physiological conditions. Biochemical assays of 5-HT binding depend on downstream signaling events and do not measure binding directly. Even measuring current evoked in voltage-clamp experiments on 5-HT_3_ receptors provides only indirect information regarding ligand binding, as the binding equilibrium is inextricably coupled to the gating process.

An ideal in vitro ligand binding assay would meet five criteria: (1) it would allow for the measurement of 5-HT binding in real time, (2) be non-destructive and chemically reversible, (3) permit measurements to be made on intact, functional channels heterologously expressed in living cells, (4) have a short distance dependance so binding can be measured site-specifically, and (5) be compatible with high-throughput biochemical assays and (for 5-HT_3_ receptors) voltage clamp. In previous work, we developed such an assay to measure nucleotide binding to ATP-sensitive K^+^ channels (K_ATP_) [9,10]. This assay involved expression (in HEK293T cells) of channels labeled with the fluorescent, non-canonical amino acid l-3-(6-acetylnaphthalen-2-ylamino)-2-aminopropanoic acid (ANAP, Figure 1A) [11]. ANAP can be introduced site-specifically into any site on a protein using amber stop-codon suppression. We previously used Förster resonance energy transfer (FRET) between ANAP and fluorescent ATP derivatives to measure binding to K_ATP_ [9,10]. The strong distance-dependence of FRET allowed us to measure binding in a site-specific fashion (i.e., we were able to discriminate binding between three different classes on nucleotide-binding site on K_ATP_).

We now report an in vitro method that allows measurement of 5-HT binding to its receptors that satisfies the five criteria stated above. Reasoning that Prodan (structurally analogous to the side chain of ANAP) fluorescence is quenched by tryptophan and that 5-HT is derived from tryptophan, we first show that 5-HT, as well as dopamine (DA), quench ANAP in solution. We further investigate this process using density functional theory, time-resolved photoluminescence and transient absorption spectroscopies to demonstrate that quenching occurs via a short-range electron transfer process that reduces ANAP and recovers reversibly via back-electron transfer. Finally, we expressed ANAP-labeled 5-HT_3A_ receptors in HEK293T cells and used quenching to measure 5-HT binding to channels in the plasma membrane of living cells. These results point to the broad applicability of ANAP for directly interrogating the binding of 5-HT to relevant receptors in vitro, allowing for an expanded understanding of the biophysics of transmitter binding and receptor activation that underly their physiological signaling roles, as well as how dysfunction in that signaling produces pathological states.

## 2. Results

### 2.1. Steady-State Quenching of ANAP by 5-HT and DA in Solution

The fluorophore Prodan, structurally analogous to the side chain of ANAP, is quenched by tryptophan in micelles as well as when both molecules are incorporated into cytochrome P450 [12]. Given its similarity to Prodan and its utility as a site-specific protein label in vitro, we sought to determine whether ANAP could be quenched by 5-HT, a neurotransmitter derived from tryptophan. We first measured the quenching of ANAP by 5-HT in water. At high concentrations (10–50 mM), 5-HT caused a blue shift in ANAP fluorescence (from 499 nm to 487 nm) as well as a 53% reduction in the peak intensity (Appendix A). The blue shift may have been the result of ANAP’s reported environmental sensitivity, but may also have resulted from pH changes in unbuffered solutions [11]. Therefore, we measured the quenching by 50 mM 5-HT in phosphate-buffered saline (PBS) as well as the buffer we used for cell-based assays (recording buffer). 50 mM 5-HT quenched the ANAP fluorescence by 59% in phosphate-buffered saline (Appendix A) and 60% in recording buffer (Appendix A). We did not observe any change in the peak emission wavelength for ANAP upon 5-HT addition in buffered solutions, suggesting the shift we observed in water may have been pH dependent. Regardless, to reduce any solvatochromic shifts and focus solely on quenching, we measured the spectrum of 20 µM ANAP at varying 5-HT concentrations in DMSO, where the peak emission wavelength was 451 nm. This greatly reduced any additional shift in peak fluorescence induced by 5-HT binding.

In DMSO, 5-HT greatly attenuated the peak ANAP fluorescence in a concentration-dependent manner (Figure 1B). Quenching was evident at concentrations of 5 mM 5-HT and above. A Stern–Volmer plot of the quenching data was well fit with a straight line (Figure 1C; K_sv_ = 13.3 M^−1^, intercept = 0.98, R^2^ = 0.998). The linear slope and concentration-dependence are both consistent with collisional quenching. 5-HT absorbs light over the concentration range at which ANAP is excited and emits light, which was evident at high concentrations (Appendix A). Therefore, the absorbance spectrum of each solution was measured and used to correct for the inner-filter effect. Corrected data for 5-HT quenching are shown in Appendix A. Whereas there was some deviation from linearity after applying this correction, the resulting Stern–Volmer plot was still well fit with a straight line (Appendix A; K_sv_ = 4.31 M^−1^, intercept = 0.98, R^2^ = 0.977). At low 5-HT concentrations, we observed a slight increase in fluorescence rather than quenching, particularly in the corrected data (Appendix A). We attribute this change to a residual solvatochromic effect, such as the one we observed in water and the one reported by Chatterjee et al., where an increase in the apparent quantum yield of ANAP accompanying the blue shift in emission [11]. We observed a small, reproducible shift in the peak wavelength for ANAP emission (from 451 nm to 447 nm) in the presence of 100 mM 5-HT in DMSO, again consistent with ANAP’s reported environmental sensitivity (Figure 1B).

In addition to its significant role as a modulatory neurotransmitter, DA is a weak partial agonist of 5-HT_3_-type receptors [13]. Therefore, we tested DA’s ability to quench ANAP fluorescence in solution. Similarly to 5-HT, DA quenched ANAP fluorescence in a concentration-dependent manner (Figure 1D). The Stern–Volmer plot for DA quenching was linear (Figure 1E; K_sv_ = 2.69 M^−1^, intercept = 0.99, R^2^ = 0.975). The shallower slope compared to 5-HT implies a less efficient quenching process for DA [14]. DA does not absorb light as strongly as 5-HT over the wavelength range at which ANAP absorbs and emits (Appendix A). Nevertheless, we corrected for the inner filter affect using the absorbance spectra measured for each solution (Appendix A). The Stern–Volmer plot for corrected data was still linear and produced a fit with similar parameters (Appendix A; K_sv_ = 2.70 M^−1^, intercept = 0.98, R^2^ = 0.951). As for 5-HT, we observed a small increase in ANAP fluorescence at low quencher concentrations. Again, we attribute this increase to the reported environmental sensitivity of ANAP [11]. Quenching of ANAP fluorescence by 5-HT and DA is consistent with criterion 1 of the 5 that we laid out in the Introduction, suggesting that ANAP can serve as a useful 5-HT detector.

### 2.2. Time-Resolved Spectroscopies

Excited-state behaviors of ANAP and 5-HT were further interrogated using time-dependent spectroscopies. Time-resolved photoluminescence measures light emitted per unit time over sub-nanosecond time bins, revealing the fluorescent lifetimes of excited species following pulsed laser excitation. TRPL measurements at 480 nm following excitation with a 355 nm pump laser revealed fluorescence lifetimes shorter than the laser pulse full width at half max (FWHM) but within range of our detector time resolution (400 ps), producing data most accurately fit by a convolution of the instrument response function (IRF) and a single exponential decay (Figure 2 and Appendix A). This relatively low-parameter model provided good agreement with data but did result in some meaningful residual values at short times due to the model neglecting the picosecond-timescale vibrational cooling that modulates the onset of fluorescent emission (Appendix A). Lifetimes were 2.231 ± 0.001 ns for 5-HT and 2.776 ± 0.002 ns for ANAP, with the latter exhibiting the expected qualitatively much larger molar absorptivity and emission. The uncertainties represent a lower limit, as they address only fit error and do not encompass additional contributions to timescale uncertainty due to other factors including concentration and instrument time resolution; the 400 ps time bins used in this experiment suggest that the upper bound on lifetime uncertainty is ±0.4 ns for all species. The lifetimes displayed the expected solvent-induced deviation from published values for ANAP (described by two exponential components of 1.3 ns and 3.3 ns) and 5-HT (4 ns) [15,16]. The effect of solvent on fluorescence lifetime is well-studied and our measurements in DMSO are not expected to correspond directly to those earlier studies performed in aqueous solution [17,18,19].

Our TRPL measurements of emission from a mixture of both compounds (100 nM ANAP with 10 mM 5-HT) exhibited the expected individual 5-HT and ANAP signal components but with decreased overall intensity. After subtracting the background 10 mM 5-HT fluorescence signal, the pattern of delayed ANAP emission (Figure 2) and corresponding shortened ANAP lifetime (2.14 ± 0.02 ns) is consistent with a dynamic quenching mechanism in which ANAP fluorophores must diffuse through solution to find 5-HT quenchers; ANAP molecules in close proximity to 5-HT at the moment of excitation are those that quench selectively prior to fluorescing. The high concentration of 5-HT needed to produce this effect suggests that diffusing ANAP molecules can only quench via a short-range interaction, consistent with an electron-transfer mechanism. When bound together in the context of a protein binding site, we expect this dependence on diffusion to be altered.

Transient absorption spectroscopy uses a white-light “probe” to quantify the time-dependent change in UV–visible absorbance of our system following a laser excitation “pump” pulse. TAS can thus measure the dynamics of the short-lived species that exist in solution after ANAP is quenched by 5-HT. TAS measurements required increasing fluorophore and quencher concentrations to reduce fluorescence background. Even so, this background overwhelmed the detector and resulted in artifacts for the few nanoseconds at peak irradiance. Following this artifact, a clear product signal appears and persists for several nanoseconds. To address residual electromagnetic interference (EMI; oscillating pattern) remaining in the TAS signal, we again fit with a convolution of the IRF. A single exponential function provided the best fit (superior to second-order-kinetics t^−1^ or biexponential) and indicated that this product peak fully decayed with a lifetime of 3.18 ± 0.08 ns, consistent with a chemically reversible process (Figure 3A). The slight mismatch between the EMI noise phase in the IRF and TAS data is the result of a beating effect stemming from the signal math necessary to produce the TAS data:(1)$$∆absorbance(t)=\;-l{og}_{10}\left(\frac{I_{pumped}(t)-I_{fluorescence}(t)}{I_{unpumped}(t)}\right)$$

A negative ∆OD signal (bleach) may also be observed in TAS measurements, resulting from a reduction in concentration of light-absorbing ground-state ANAP. No bleach signal was observed, though this may be attributable to the strong absorbance of the solution itself in the area around 350 nm where such bleach would be expected, resulting in a poor signal-to-noise ratio obscuring the bleach. The initial conditions were fully recovered between shots.

Further conclusions can be drawn from comparing this TAS signal to that observed from the separate solutions of ANAP and 5-HT. ANAP shows a permanent increase in absorbance following the pump pulse, indicating an irreversible change following excitation, whereas 5-HT showed no significant TAS signals. If it were the case that ANAP and 5-HT did not interact, summing the signals of separate ANAP and 5-HT solutions should reproduce the signal of the mixture; instead, the summed signal shows a permanent increase in signal in comparison with the ANAP/5-HT mixture’s nearly complete recovery (Figure 3B).

FRET or Dexter quenching mechanisms would be expected to result in the fluorophore/quencher pair being completely returned to their electrically neutral electronic ground states following the quench. Our observation of a longer-lived 3.18 ns signal is consistent with the formation of a charge-transfer state that persists in solution after the quench event has occurred [20,21]. In conjunction with DFT results (see below), we therefore attribute the observed product signal to an ANAP^•−^ radical anion resulting from reduction of ANAP by 5-HT. As we have demonstrated in earlier work, aromatic radical anions typically show visible absorbances redshifted from those of their neutral parent species due to the additional occupation of π* orbitals [22].

### 2.3. Density Functional Theory

Whereas time-resolved measurements revealed a nanosecond-lifetime excited state consistent with charge transfer (CT) as the mechanism behind the quenching of ANAP, DFT calculations provided additional insight into the directionality of the CT process (Figure 4). Comparison of the energy of a reduced/oxidized pair (e.g., ANAP^−^/5-HT^+^) to the ground-state combination (ANAP^0^/5-HT^0^) revealed the relative stability of each radical ion pair following the charge transfer event and confirmed that for all combinations of fluorophores (l-ANAP methyl ester, Prodan) and quenchers (5-HT, dopamine), the CT state was always significantly less stable (>70 kcal/mol) than the neutral ground state, consistent with the expected behavior of a reversible CT process. Prodan was included in our calculations as the side chain of ANAP is based on the structure of Prodan. DFT results also consistently showed that electron transfer from quencher to fluorophore (i.e., fluorophore reduction) resulted in a comparatively more stable CT state (73.0–77.1 kcal/mol, versus 93.2–100.5 kcal/mol for fluorophore oxidation), consistent with the expected greater stability of an electron delocalized in a larger π system [23]. Energies required to access the fluorophore oxidation pathway correspond to <300 nm photons unavailable in our experiments, whereas the energies associated with the quencher oxidation pathway are in the ~400 nm range accessible with excitation sources employed in this work. This further supports electron transfer from quencher to fluorophore as the quenching mechanism.

### 2.4. Measurement of 5-HT Binding to Intact, Functional 5-HT_3A_ Receptors

With the mechanism and dynamics of ANAP quenching by 5-HT established spectroscopically and computationally, we next expressed ANAP in the ligand binding site of 5-HT_3A_ receptors. 5-HT_3A_ is a PLGIC that is activated by direct 5-HT binding to its extracellular domain (Figure 5). 5-HT_3A_ forms functional, homomeric receptors in native and heterologous expression systems [4,24]. Typical of PLGICs, 5-HT binds in a conserved ligand-binding pocket formed at the interface of adjacent subunits, where it is coordinated by a cage of aromatic amino acids (Figure 5B). We sought to express 5-HT_3A_ receptors in which one of these aromatic amino acids (Y234) is replaced with ANAP (5-HT_3A_-Y234ANAP). We chose this site because Y234 coordinates 5-HT directly and the expected quenching interaction (via charge transfer) should only operate over very short distances. Furthermore, as tyrosine is a bulky aromatic amino acid, substitution with ANAP should be relatively conservative.

To express 5-HT_3A_-Y234ANAP channels, the gene for mouse 5-HT_3A_ was mutated to replace the triplet encoding Y234 with the amber (TAG) stop codon. When expressed alone in HEK293T cells, the resulting plasmid is expected to result in 5-HT_3A_ protein prematurely truncated at position 233 in the extracellular region preceding the first transmembrane domain. To obtain full-length, ANAP-labeled channels, cells were co-transfected with pANAP, which encodes a tRNA that recognizes the amber stop codon and a synthetase enzyme capable of charging this tRNA with ANAP. ANAP was added to the cell culture media at the time of transfection. When both plasmids and ANAP are present (Figure 6A), full-length 5-HT_3A_-Y234ANAP channels are produced. As an additional control, we inserted an mOrange protein tag in an unstructured cytoplasmic loop downstream of amino acid 234. Thus, orange fluorescence (in addition to ANAP fluorescence) indicates successful incorporation of the label and translation of full-length channels.

Figure 6A shows a series of images of HEK293T cells transfected with the plasmid encoding 5-HT_3A_-Y234ANAP. When this plasmid was co-transfected with pANAP and ANAP was included in the culture media, there was obvious fluorescence for both ANAP and mOrange, indicating the presence of full-length, ANAP-labeled channels. When ANAP was omitted from the bathing medium, we typically observed no fluorescence for either ANAP or mOrange, indicating that there was no read-through of the amber stop codon without ANAP present. When ANAP was present, but pANAP was not included in the transfections; again no mOrange fluorescence was observed. The small, diffuse ANAP fluorescence we observed under these conditions likely represents unincorporated ANAP accumulating inside cells. However, we cannot exclude the possibility that ANAP may be incorporated into other cellular proteins. Regardless, we identified cells for further experimentation based on the presence of both ANAP and mOrange fluorescence. The presence of both fluorophores implies that the signal we record is specific to labeled 5-HT_3A_ receptors.

Whereas it is necessary to place ANAP very close to the 5-HT in order to observe any short-range quenching via charge transfer, mutating amino acids so near the ligand binding site runs the risk of altering the receptor’s response to 5-HT. Thus, we used whole-cell patch clamp electrophysiology to verify that 5-HT_3A_-Y234ANAP receptors were functional and responded to 5-HT. Figure 6B shows a typical whole-cell current response to 240 µM 5-HT for an HEK293T cell expressing 5-HT_3A_-Y234ANAP. Bath application of 5-HT produced a desensitizing current typical of 5-HT_3A_. We observed an average of 41.2 ± 9.9 pA of current in response to applications of 24 µM 5-HT in 5/7 cells that we identified based on mOrange fluorescence. Thus, we can conclude that 5-HT_3A_-Y234ANAP receptors are present at the plasma membrane and maintain their response to 5-HT, despite alterations to the ligand binding site.

To measure 5-HT binding to 5-HT_3A_-Y234ANAP, we expressed the receptor in HEK293T cells and imaged them using an inverted microscope coupled to a spectrograph. This allowed us to obtain full emission spectra of ANAP-tagged channels (Figure 6C). The ANAP peak was 495 ± 5 nm, close to the value we measured when ANAP was dissolved directly in our recording buffer (492 nm; Appendix A), suggesting that ANAP at position 234 of 5-HT_3A_ was accessible to bulk water. The secondary peak at ~560 nm corresponds to mOrange (emission maximum 562 nm) [26]. Application of 5-HT gave rise to a concentration-dependent quenching of ANAP fluorescence. The apparent reduction in mOrange fluorescence was due to the overlap in ANAP and mOrange spectra. At saturating concentrations, 5-HT did not quench 100% of the ANAP fluorescence. Quenching at saturating concentrations was variable, with an average of 66% ± 7%. To compare across cells, we subtracted the fluorescence at saturating [5-HT] from each trace and then normalized to the maximum fluorescence at the lowest agonist concentration (Figure 6D). The resulting concentration–response relationship was well-fit with a modified Hill equation with half-maximal quenching at 594 nM 5-HT and a slope factor of −0.46. The inability of 5-HT to quench 100% of the fluorescence may be a result of the distance between ANAP and 5-HT or the geometry of the binding site. However, it is possible that the unquenched fluorescence signal arose from ANAP-tagged trafficking intermediates in the ER/Golgi that would not be accessible to 5-HT applied extracellularly. To test this hypothesis, we repeated the quenching experiment in unroofed plasma membranes obtained by blotting adherent cells with filter paper (Appendix A). Blotting removes intracellular organelles, thus reducing any background from trafficking intermediates. We observed a similar concentration–response relationship for quenching in unroofed membranes (EC_50_ = 160 nM, slope factor −0.49). The difference in apparent affinity between whole cells and unroofed membranes may result from the loss of important cytoplasmic factors during the unroofing process. As we observed in intact cells, we did not obtain 100% quenching in unroofed membranes at saturating 5-HT concentrations. Therefore, the inability of 5-HT to completely quench likely reflects the geometry of the binding site or photophysics of the ANAP/5-HT interaction, not an inaccessible population of channels.

## 3. Discussion

Based on our results, we propose that ANAP can be used to detect 5-HT binding to its receptors in vitro. Our assay meets all five criteria set out in the Introduction. By using fluorescence, we measured binding in real time to intact, functional channels heterologously expressed in living cells. Based on our time-resolved fluorescence and transient absorption measurements, we observed a reversible quenching process that recovers with a time constant of 3.2 ns, likely due to back-electron transfer from the reduced fluorophore [27]. This recovery time for the intermolecular process represents an upper bound to back-electron transfer within a ligand binding site and is nonetheless much faster than the expected time course of 5-HT dissociation from its receptors. Thus, 5-HT can quench excited ANAP molecules without any irreversible bleaching of the fluorophore. Charge transfer is very sensitive to distance (requiring collision/contact between the fluorophore and quencher). Therefore, ANAP/5-HT CT allows for detection of binding very near to the site of ANAP incorporation and would not be sensitive to 5-HT bound elsewhere. Replacing Y234 in the 5-HT binding site of 5-HT_3A_ still allowed for binding of 5-HT and channel activation. Thus, with a sufficient signal-to-noise ratio, this binding assay is compatible with simultaneous measurements of channel current so that ligand occupancy can be directly correlated with changes in function (i.e., activation, desensitization). Finally, our fluorescence-based assay can be readily scaled for use in a 96- or 384-well format to enable medium/high throughput data acquisition.

We used ANAP-tagged 5-HT_3A_ receptors in our assay because, as 5-HT_3A_ is an ion channel, we could record currents to verify that our ANAP-tagged receptors were functional, expressed in the plasma membrane, and able to respond to 5-HT. However, this technique should be easily adapted for use in G-protein-coupled 5-HT receptors and 5-HT transporters. Based on our steady-state measurements, ANAP may also be a promising DA detector when incorporated into receptors. DA is a partial agonist of 5-HT_3A_ and comparing current responses and binding measurements between 5-HT and DA may be a promising way to explore the mechanism of partial agonism in these channels. Furthermore, ANAP can be incorporated into the binding site of DA receptors to help illuminate their biological role.

In solution, very high concentrations of both 5-HT and DA were required to quench ANAP (Figure 1). This is consistent with dynamic quenching via an electron transfer process. As such processes require the fluorophore and quencher to be within contact distance, quenching in solution would only be expected to happen at very high concentrations where donor-quencher collisions are more probable (assuming no high-affinity binding between the pair). The concentration range over which we observed quenching in solution far exceeds physiological concentrations of 5-HT. In the gut lumen, the 5-HT concentration is around 0.01 mg/L (around 57 nM) [28]. In whole blood, the 5-HT concentration is 774 nM [29]. Varying concentrations (0.1–65 nM) have been reported for 5-HT in the brain [30], although the 5-HT concentration in the synapse can rise as high as 2 µM in response to repeated stimulation [31].

In the context of ANAP-labeled receptors, however, we observed quenching at much lower concentrations than in solution (Figure 6). The shift in concentration dependence is expected because 5-HT binds with relatively high affinity to 5-HT_3A_ positioning the quencher within contact distance of ANAP (i.e., quenching is no longer dependent on random collisions in solution). The values we obtained from fits to our 5-HT binding data (EC_50_ 594 nM) were in the range of EC_50_ values obtained from functional measurements of recombinant 5-HT_3A_ activity. The EC_50_ for activation of 5-HT_3A_ in HEK293T cells measured with Ca^2+^ imaging was 348 nM [32]. In two-microelectrode studies in *Xenopus* oocytes half-maximal current activation occurred at 3.54 µM [4]. The differences in the values we obtained may be the result of mutating the 5-HT binding site. Alternatively, the differences between our measurement and reported values may reflect a difference in experimental approach or expression system. Lastly, the discrepancy between our EC_50_ value and published values may represent state-dependent 5-HT binding. Based on the time course of the channel currents we measured (Figure 6B) and the duration of our 5-HT applications, we assume that we measured the apparent 5-HT affinity for desensitized receptors. Regardless, one should exercise caution when interpreting EC_50_ values as reflecting any true binding affinity. Even for assays that measure “binding” directly, any values reported reflect not only the binding equilibrium, but any coupled conformational equilibria in the channel complex [33].

As presented, our method offers great promise for understanding the relationship between binding of transmitters like 5-HT and DA and receptor activation. However, there are several points of caution worth considering. Our primary difficulty is that, even at saturating concentrations of 5-HT, maximal quenching of 5-HT_3A_-Y234ANAP was only around 66% and variable. Whereas the signal-to-noise ratio allowed us to generate reproducible binding curves in whole cells and unroofed membranes, more robust and more reproducible quenching at saturating 5-HT concentrations is desirable. We believe that the inability of saturating concentrations of 5-HT to completely quench ANAP in the context of 5-HT_3A_-Y234ANAP receptors may reflect the very short distance dependence of the electron transfer process that mediates quenching or that the donor and acceptor may be bound in an unfavorable orientation for efficient electron transfer. Replacement of other aromatic amino acids in the binding site with ANAP may produce better results.

In solution, low concentrations of DA and 5-HT produced an apparent increase in ANAP fluorescence. We attribute this to the reported environmental sensitivity of ANAP fluorescence [11]. Whereas we did not observe a consistent increase in fluorescence when 5-HT bound to ANAP-tagged 5-HT_3A_, this solvatochromic effect could introduce error into our binding measurements and may partially account for the inability of 5-HT to quench ANAP 100% at saturating concentrations (Figure 1 and Figure 6D).

We used long (10–30 s) applications of 5-HT in our experiments and 10 s exposure times for our spectra. On this time scale, 5-HT_3A_-Y234ANAP receptors were likely in the desensitized state. To capture shorter-lived gating intermediates, faster solution exchange and shorter exposure times are necessary. The latter will be easier to achieve in preparations with a greater signal-to-noise ratio. Alternatively, mutations that reduce sensitization or allosteric modulators that delay it may help explore binding to the open state.

Our method is distinguished by the ability to measure 5-HT binding directly to heterologously expressed receptors in living cells/cell membranes and in real time. Historically, radioligand binding assays/competition assays have been used to measure ligand binding affinity. However, such assays often require long incubation times (up to 48 h) to achieve equilibrium, and can suffer from artifacts due to ligand depletion, non-specific ligand binding, or contamination from unbound ligand [34,35]. Furthermore, the extracted parameters are highly sensitive to experimental conditions [35,36]. Our approach mitigates some of these concerns. The signal in our assay arises from ANAP-labeled receptors, so any unbound or non-specifically bound 5-HT would not be detected. As we perfuse our cells with a constant 5-HT concentration, ligand depletion artifacts should be non-existent. Furthermore, our assay for 5-HT_3A_ can be paired with measurements of channel currents at varying 5-HT concentrations to verify that our parameter estimates are accurate over a time scale relevant to channel gating and consistent with channel activity under physiological conditions.

Numerous other fluorescence-based assays have been developed to infer 5-HT binding to its receptors. Henke et al. [37] developed FFN246, a fluorescent, false neurotransmitter that binds to the 5-HT transporters SERT (serotonin transporter) and VMAT (vesicular monoamine transporter). This compound could be used to measure 5-HT affinity in competition assays. However, even in in vitro assays, the indiscriminate binding to multiple receptor types may complicate interpretation of results. In the Tango method ligand binding to 5-HT_2C_ is inferred indirectly from downstream GFP expression [38]. G-protein coupled 5-HT receptors can also be engineered with circularly permuted GFP molecules inserted between TM5 and TM6 so that ligand binding can be inferred from the subsequent conformational change [39,40,41]. Cell-based neurotransmitter fluorescent-engineered reporters (CNiFERs) have also been designed with a Ca^2+^ sensitive moiety attached to 5-HT_3A_ to measure the Ca^2+^ influx subsequent to channel opening [42]. For 5-HT_3_-family receptors, current measurements are also used to assess ligand binding. However, while all of these assays are useful, none of them measure ligand binding directly. By placing ANAP in the 5-HT binding site, our approach provides gives a direct readout of receptor occupancy. As mentioned above, one must exercise caution when interpreting parameters extracted from any binding assay, including ours, performed on proteins for which binding is coupled to some conformational change (like receptor activation). For such proteins, any apparent affinities extracted reflect not only the true binding affinity, but any equilibria coupled to binding [33]. However, when performed simultaneously with measurements of channel currents, assays like ours can be used to extract meaningful parameters regarding ligand binding affinity and coupling between binding and receptor activation [10].

Whereas our method is well suited to in vitro studies in cultured cells, it would not be readily adaptable to studies in vivo. To our knowledge, there are no studies that show expression of ANAP-tagged proteins in whole organisms (other than single-celled organisms) [43]. Furthermore, as both 5-HT and DA quench ANAP fluorescence, it would be difficult to discriminate between the two transmitters outside of a cell-based assay in which the experimenter has control over the extracellular milieu. Zhao and Piatkevich provide a complete discussion regarding in vivo 5-HT detection [30].

## 4. Materials and Methods

### 4.1. UV/Vis Absorbance and Steady-State Fluorometry

Stock solutions of ANAP (ASIS ChemInc., Waltham, MA, USA) were prepared at 20 mM in DMSO and diluted into DMSO, water, or buffered solutions. 5-HT and DA stocks were prepared at 200 mM in water or DMSO. Stocks were stored frozen at −20 °C. To reduce the impact of oxidation, all 5-HT and DA solutions were prepared fresh the day before experiments and kept frozen until the time of measurement. The absorbance of each ANAP/5-HT or ANAP/DA solution was measured using a Red Tide UBS650U spectrometer (Ocean Optics; Orlando, FL, USA) and LoggerPro software 3.16.2 (Vernier; Beaverton, OR, USA). 5-HT and DA blank solutions were measured separately. All spectra were acquired using a low-volume quartz cuvette (Azzota Corp.; Claymont, DE, USA).

The fluorescence emission spectrum of 20 µM ANAP in water or DMSO was measured in different concentrations of 5-HT or DA, as indicated. Spectra were also acquired in phosphate-buffered saline (PBS; 0.01 M phosphate buffer, 2.7 mM KCl, 137 mM NaCl, pH 7.4) and the recording buffer used for cell-based experiments (140 mM NaCl, 2.8 mM KCl, 2 mM MgCl_2_, 1 mM CaCl_2_, and 10 mM HEPES, pH 7.4 with N-methyl-D-glucamine). Spectra were acquired using a Hitachi F-7000 fluorescence spectrophotometer (Hitachi High-Tech, Tokyo, Japan) and FL Solutions 4.2 software. The samples were excited at 370 nm and emission was measured between 380 nm and 600 nm. Slit widths were 5 nm on the excitation and emission side and the PMT was set to 400–700 V. Spectra were blank subtracted using emission spectra of 5-HT/DA solutions without ANAP. DA contributed very little to the fluorescence background, whereas 5-HT accounted for 15.2% of the total fluorescence signal at the peak wavelength for ANAP fluorescence (448 nm) measured for 20 µM ANAP plus 100 mM 5-HT in DMSO.

Emission spectra were corrected for the inner-filter effect (as indicated) using absorbance values measured as above and the following equation(2)$$F=F_{obs\;}\times 10^{\frac{\left({0.2*OD}_{ex}+{OD}_{em}\right)}{2}}$$
where F_obs_ is the measured emission, OD_ex_ is the optical density at the excitation wavelength (370 nm) and OD_em_ is the optical density at each emission wavelength. The 0.2 term corrects for the low-volume cuvette, which had a short, 0.2 cm path oriented toward the excitation light. Before correction, UV/Vis spectra were smoothed in JMP (JMP Statistical Discovery LLC, Cary, NC, USA) using local kernel smoothing with a local bandwidth of 0.25 nm.

Stern–Volmer plots were fit with a linear equation of the form(3)$$\frac{F_{0}}{F}={F_{intercept}+K}_{sv}\times [quencher]$$
where F is the averaged peak fluorescence between 445 nm and 455 nm for each solution, F_0_ is the average fluorescence in the absence of quencher, and K_sv_ is the Stern–Volmer constant, reflecting the quenching efficiency. F_intercept_ was the value extrapolated from the linear fit to zero [quencher]. This value is expected to be one and was found to be close to one in our fits.

### 4.2. Density Functional Theory Calculations

The relative stability of fluorophore/quencher electron transfer pairs were computed using density functional theory (DFT). Q-Chem 5.2 was used to compute neutral singlet structures, as well as reduced and oxidized doublet structures, of each compound using the ωB97XD density functional and def2-TZVPP basis set and a polarizable continuum model (ε = 80) [44,45]. Doublet calculations were restricted to prevent spin contamination [46]. Following initial structure optimizations, harmonic frequency analysis confirmed that each structure represented a true local minimum (no imaginary frequencies) and provided values to determine ∆H and ∆S corrections at 310 K for the calculation of theoretical ∆G values [47]. Thermodynamics comparisons and plotting were performed using Python 3.9.13 with numpy 1.26.2 and matplotlib 3.8.0.

### 4.3. Time-Resolved Spectroscopies

Time-resolved photoluminescence (TRPL) and transient absorption spectroscopy (TAS) measurements were performed using an Edinburgh Instruments LP980 Transient Absorption Spectrometer (Edinburgh Instruments Ltd, Livingston Village, UK) with a Litron Nano (Litron Lasers, Rugby, UK) 5 Hz pump laser tripled to produce 355 nm, 8 ns pulses (FWHM). Both TRPL and TAS data were taken with the LP980’s PMT detector configured in its highest time-resolution mode, corresponding to 400 ps time bins. TRPL measurements were performed with the laser attenuator set to 10 mJ/pulse on solutions of 10 nM ANAP or 10 mM 5-HT in DMSO; emission was measured at 480 nm for 64 averages. For TAS measurements, laser power was lowered to 1 mJ/pulse and concentrations were increased to 1.0 mM ANAP and 10 mM 5-HT and were followed via ANAP radical anion peak at 440 nm for 500 averages (made possible by the complete recovery of signals between shots). Data were fit to a convolution of the instrument response function (IRF; measurement of elastically scattered laser pulse after passing through ND = 3 filter) and exponential decay function(s) using Edinburgh Instruments’ L900 version 9.4.4 software [48]. Additional analysis and plotting were performed using Python (see above).

### 4.4. Molecular Cloning

Mouse 5-HT_3A_ in pICherryNeo was a gift from Dr. Sudha Chakrapani. pCGFP_EU was a gift from Dr. Eric Gouaux [49]. mOrange-N1 was a gift from Michael Davidson (Addgene plasmid # 54499, Watertown, MA, USA; http://n2t.net/addgene:54499, accessed on 1 October 2025; RRID: Addgene_54499) [50]. pANAP was a gift from Peter Schultz (Addgene plasmid # 48696; http://n2t.net/addgene:48696, accessed on 1 October 2025; RRID: Addgene_48696) [11].

As a preliminary step, the GFP insert from pCGFP_EU was cloned into the 5-HT_3A_ sequence in pICherryNeo at a position between amino acids D382 and S384 in the putatively unstructured loop between transmembrane domains 3 and 4 [25]. We used PCR (QuickChange Kit; Agilent Technologies; Santa Clara, CA, USA) to introduce new restriction sites into the 5-HT_3A_ sequence and inserted the GFP tag using standard subcloning techniques. We subsequently replaced this GFP tag with mOrange because the emission peak for mOrange was farther from the expected ANAP peak. Stop-codon mutations were introduced using QuickChange. All constructs were verified by direct sequencing at Azenta GeneWiz (South Plainfield, NJ, USA).

### 4.5. Cell Culture

HEK-293T cells were purchased from ATCC (Manassas, VA, USA). Frozen stocks were prepared during early passages and stored in liquid nitrogen. Cells were verified by ATCC. We performed no additional verification or mycoplasma testing. Cells were grown in T-75 flasks (Grenier Bio-One; Monroe, NC, USA) in Dulbecco’s Modified Eagle Medium (DMEM; Gibco; Grand Island, NY, USA) supplemented with 10% fetal bovine serum (FBS; Corning; Tewksbury, MA, USA), 100 U/mL penicillin, and 100 µg/mL streptomycin (Gibco) at 37 °C in a 5% CO_2_/95% air atmosphere.

### 4.6. Expression of ANAP-Tagged Proteins

ANAP-tagged proteins were obtained as previously described [9,11,43]. Cells were co-transfected with 1 µg of 5-HT_3A_ plasmid with an amber stop codon (TAG) replacing the codon corresponding to amino acid Y234 and a downstream mOrange tag, as well as pANAP, which encodes a tRNA with the appropriate anticodon (CUA) to recognize the amber stop codon and a synthetase enzyme capable of charging the tRNA with ANAP. Cells were transfected using TransIT transfection reagent (Mirus; Madison, WI, USA) at a ratio of 3 µL/µg of DNA. Just prior to transfection, the culture medium was replaced with DMEM supplemented with FBS, penicillin/streptomycin, and 20 µM ANAP methyl ester (AsisChem Inc.; Waltham, MA, USA). Experiments were typically performed 2 days post-transfection.

### 4.7. Electrophysiology

Currents were recorded in the whole-cell patch clamp configuration. Data were acquired using a Sutter IPA amplifier controlled by SutterPatch 2.1.1 (Igor Pro 8.0.4.2) software (Sutter Instruments; Novato, CA, USA). Data were digitized at 10 kHz and low-pass filtered at 1 kHz. Patch pipettes were pulled from borosilicate glass (World Precision Instruments 1B150F-4) to a resistance of 8–10 MΩ. The pipette solution contained 140 mM KCl, 1.2 mM MgCl_2_, 2.6 mM CaCl_2_, and 10 mM HEPES, pH 7.4 with N-methyl-D-glucamine. The bath solution (recording buffer) contained 140 mM NaCl, 2.8 mM KCl, 2 mM MgCl_2_, 1 mM CaCl_2_, and 10 mM HEPES, pH 7.4 with N-methyl-D-glucamine. Bath solution was exchanged using a MiniStar peristaltic pump (World Precision Instruments). 5-HT was applied in bath solution (supplemented with 5 mM glutathione) using a pressure-driven SmartSquirt Micro-Perfusion System (AutoMate Scientific; Berkeley, CA, USA).

### 4.8. Unroofing HEK Cells for Imaging

Cells were unroofed to obtain isolated plasma membranes [51,52]. A small fragment of glass coverslip with adherent HEK293T cells expressing labeled channels was broken off using jeweler’s forceps. The fragment was dipped in a solution of 0.1% poly-l-lysine for 3 × 10 s. Fragments were blotted (cell-side down) on a clean piece of filter paper (Cytiva Whatman Grade 1; Wilmington, DE, USA), leaving behind adherent plasma membrane fragments.

### 4.9. Microscopy/Spectroscopy in Living Cells/Cell Fragments

For experiments, cells were plated in 35 mm dishes (Thermo Scientific; Waltham, MA, USA) on glass coverslips (#1 thickness; Chemglass Life Sciences; Vineland, NJ, USA) or on FluoroDishes (World Precision Instruments; Sarasota, FL, USA) a day before transfection. Cells were imaged in the same bath solution used for patch clamp using a Nikon TE2000-U (Melville, NY, USA) or Ti2E microscope equipped with a 60× water-immersion objective (Nikon Plan Apo VC, Melville, NY, USA). ANAP was excited using a ThorLabs LED source (LED4D067; Newton, NJ, USA) with a center wavelength of 385 nm. For imaging ANAP, the filter set contained a 390/18 nm band-pass filter (MF390-18, ThorLabs), an MD416 dichroic mirror (ThorLabs) and a 470/40 nm band-pass filter (MF479-40, ThorLabs). To obtain spectra, the emission filter was replaced with a 400 nm long-pass filter (FEL0400, ThorLabs). To image the mOrange tag, the same light source was used with a broad LED centered at 565 nm. A 530-30 band-pass filter was used for excitation (Chroma; Bellows Falls, VT, USA) with a T550lpxr dichroic mirror (Chroma) and 575/50 band pass emission filter (Chroma).

Images and spectra were collected using an IsoPlane81 camera/spectrograph (Teledyne Princeton Instruments; Princeton, NJ, USA) or IsoPlane160/PIXIS 256 camera (Teledyne Princeton Instruments). Image/spectrum acquisition was automated with trigger pulses from a Sutter IPA amplifier controlled by SutterPatch software (Sutter Instruments). Exposure times were typically 10 s. Photobleaching artifacts were corrected as previously described [9,51]. Briefly, five images were acquired in the absence of 5-HT, the peak intensity of the ANAP spectrum was plotted as a function of exposure time, and the intensity decay was fit with a single exponential. The resulting fit curve was used to correct subsequent exposures. For quenching experiments, 5-HT in bath solution (or bath solution +5 mM glutathione in unroofed experiments) was applied using a SmartSquirt perfusion system controlled by SutterPatch. Wash times were 10–30 s to ensure that ligand binding reached a steady state. The FluoroDish was perfused with bath solution using a MiniStar peristaltic pump. Solutions were diluted from frozen stocks just prior to the experiment and stored on ice.

*Data analysis.* Spectra and chromatograms were analyzed and plotted using custom code written in R version 4.5.0, with plotly 4.10.4, hyperSpec 0.100.3, beepr 2.0, ggplot2 3.5.1, gridExtra 2.3, and xlsx 0.6.5 packages [53,54,55,56,57,58,59]. Time-resolved spectroscopic data were analyzed using custom code in Python 3.9.13. Images were analyzed and displayed using Fiji (ImageJ 1.54P) [60]. Additional plots, and curve fits were carried out using JMP 2025 JMP^®^ (version 18.2.2; JMP Statistical Discovery LLC, Cary, NC, USA).

To compare concentration–response relationships between cells/membranes, peak fluorescence from bleach-corrected spectra from each cell/membrane were normalized by subtracting the minimum fluorescence at saturating concentrations and dividing by the peak at the lowest applied 5-HT concentration (where no appreciable quenching was evident). Concentration–response data were fit with curve of the form(4)$$\frac{\left(F-F_{min}\right)}{\left(F_{max}-F_{min}\right)}=\frac{1}{1+10^{h*((\log{EC}_{50}-\log\left[5-HT\right])}}$$
where [5-HT] is the concentration of 5-HT, F is fluorescence, F_max_ is the fluorescence intensity at the lowest [5-HT] concentration (where we observed no appreciable quenching), F_min_ is the fluorescence intensity at the highest [5-HT], and *EC*_50_ is the concentration that produced half-maximal quenching.

### 4.10. Data Presentation and Statistics

For solution spectra, data are displayed as the average of three samples (two for 20 µM ANAP in DMSO). For data from cells and unroofed membranes, all of the individual data points were displayed as well as the mean values. Errors are displayed as mean ± SEM. *n* values and fit parameters are included in the figure legends. No power analyses were conducted prior to our experiments to determine the number of repetitions. Protein structures were displayed using ChimeraX 1.10rc202506130232 developed by the Resource for Biocomputing, Visualization, and Informatics at the University of California, San Francisco with support from National Institutes of Health RO1-GM129325 and the Office of Cyber Infrastructure and Computational Biology, National Institute of Allergy and Infectious Disease [61]. Chemical structures were created and formula masses predicted using ChemDraw 20.0 (Revvity Signals Software; Waltham, MA, USA).

### 4.11. Data/Code Availability

The original contributions presented in this study are included in the article/Appendix A. Further inquiries can be directed to the corresponding author.

### 4.12. Materials

Unless otherwise noted, all chemicals were obtained from Sigma-Aldrich (St. Louis, MO, USA). Restriction enzymes were purchased from ThermoFisher and New England Biolabs, Inc. (Ipswich, MA, USA).

## 5. Conclusions

ANAP, with its sensitivity to both 5-HT and DA, shows great promise as a direct probe of transmitter/receptor interactions in heterologous expression systems. ANAP is easily inserted co-translationally into any site on a target receptor using amber stop-codon suppression. As ANAP insertion produced functional channels future experiments will allow us to explore the relationship between ligand binding and channel gating for both the full agonist 5-HT and the partial agonist DA. In our previous work, simultaneous measurements of ligand binding and changes in current allowed us to constrain K_ATP_ gating models and derive meaningful parameters regarding true ligand binding affinity and the energetic coupling between binding and activation [10]. This approach may also be used to understand how co-assembly of 5-HT_3A_ with other subunits (5-HT_3B_ through 5-HT_3E_) affects this process as well as how mutations associated with diseases like IBS, drug abuse disorder, and schizophrenia can affect the channel’s biophysical properties. Finally, this technique can be readily adapted to study other 5-HT and DA receptors and transporters, offering a real-time window into their structure and activity.

## Figures and Tables

**Figure 1 ijms-26-10815-f001:**
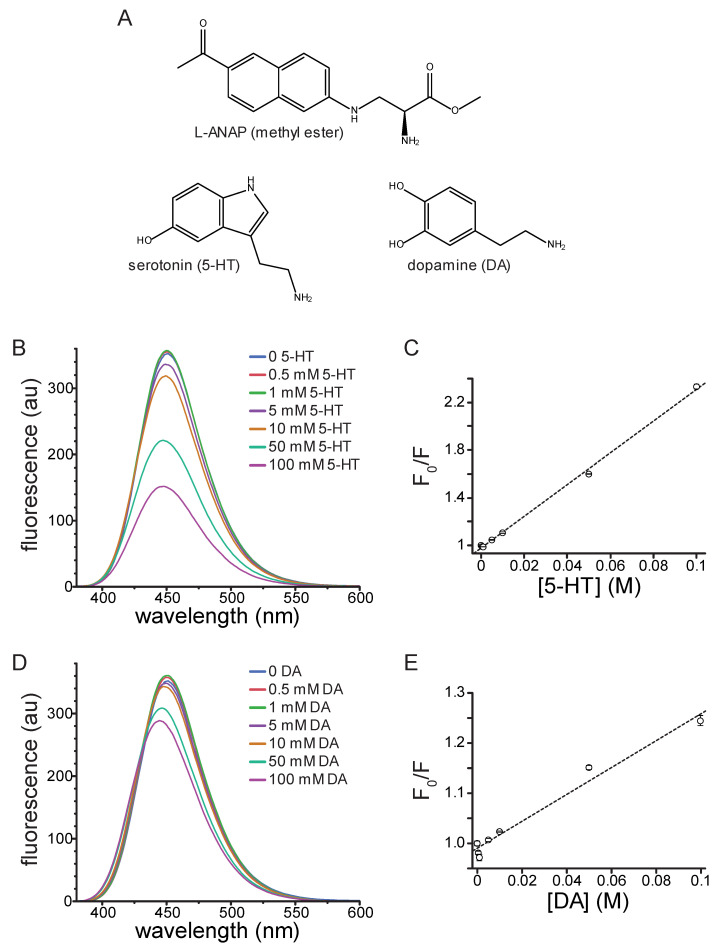
Quenching of ANAP by 5-HT and DA in solution. (**A**). Molecular structures of ANAP, 5-HT, and DA. (**B**). Emission spectra (excitation 370 nm) of 20 µM ANAP in DMSO in the absence and presence of 5-HT. (**C**). Stern–Volmer plot for 5-HT quenching of ANAP. The relationship was fit with a straight line with an intercept of 0.98 and a slope (K_sv_) of 13.3 M^−1^ (R^2^ = 0.998). *n* = 3. (**D**). Emission spectra (excitation 370 nm) of 20 µM ANAP in DMSO in the absence and presence of DA. (**E**). Stern–Volmer plot for DA quenching of ANAP. The relationship was fit with a straight line with an intercept of 0.99 and a slope (K_sv_) of 2.69 M^−1^ (R^2^ = 0.975). *n* = 3.

**Figure 2 ijms-26-10815-f002:**
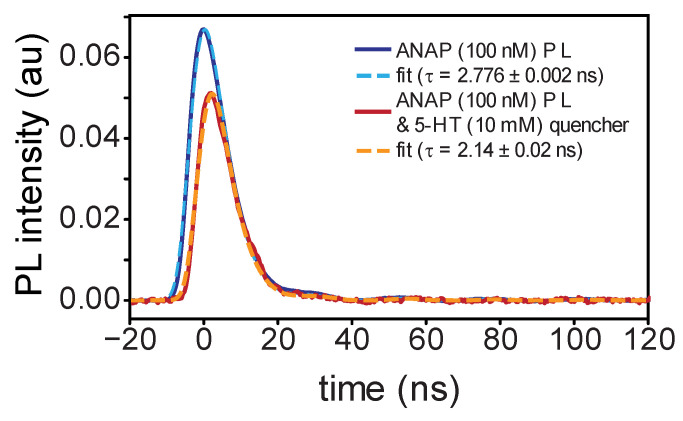
Time-resolved photoluminescence measurements of l-ANAP methyl ester in DMSO with (blue) and without (red) 5-HT present; dashed lines indicate single-exponential fit from convolution with IRF. Delayed onset of emission in the presence of 5-HT is consistent with dynamic quenching.

**Figure 3 ijms-26-10815-f003:**
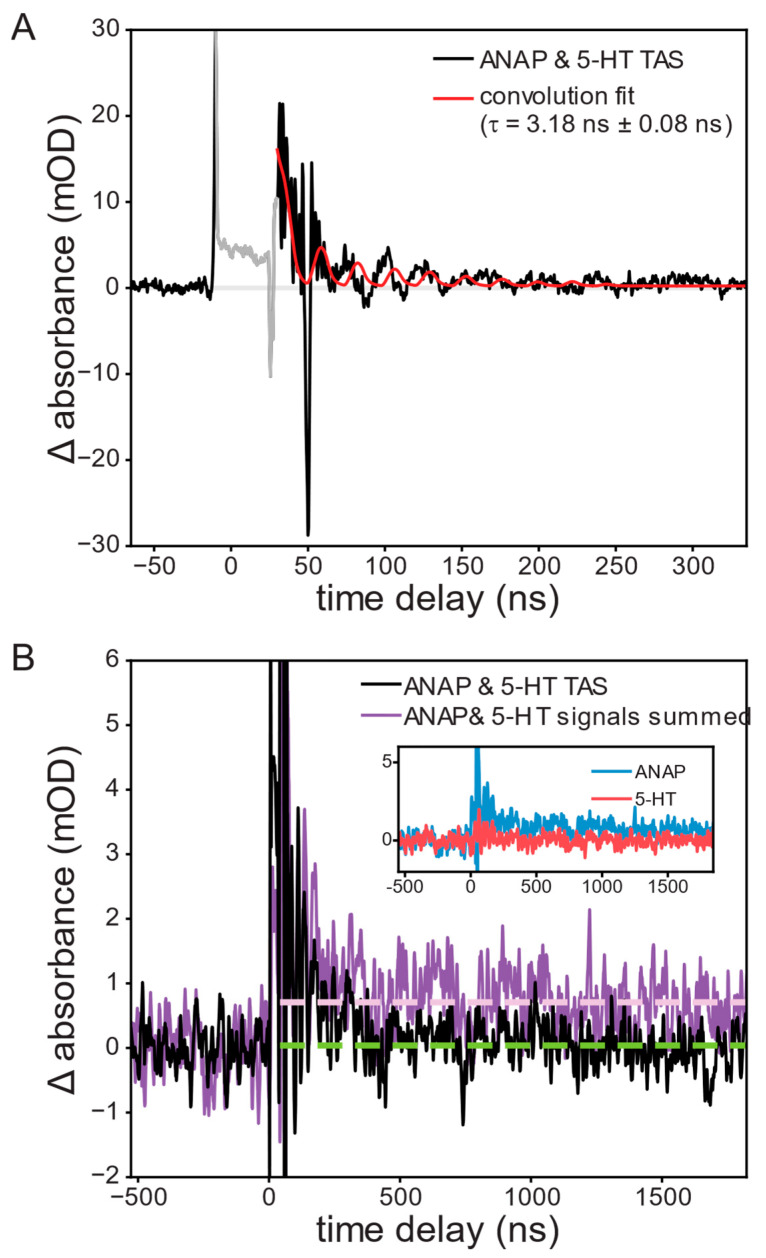
Transient absorption of a 1.0 mM l-ANAP methyl ester and 10 mM 5-HT solution at 440 nm following excitation at 355 nm, showing (**A**) the appearance and recovery of a short-lived product peak attributed to reduced ANAP; convolution fit lifetime of the resulting recovery was 3.18 ± 0.08 ns. (**B**) ANAP and 5-HT mixture (black; with 21-point second-order Savitzky–Golay smoothing) exhibits complete recovery (green dashed line) while the sum of signals from the two separate solutions (violet) shows a long-lived transient signal (violet dashed line).

**Figure 4 ijms-26-10815-f004:**
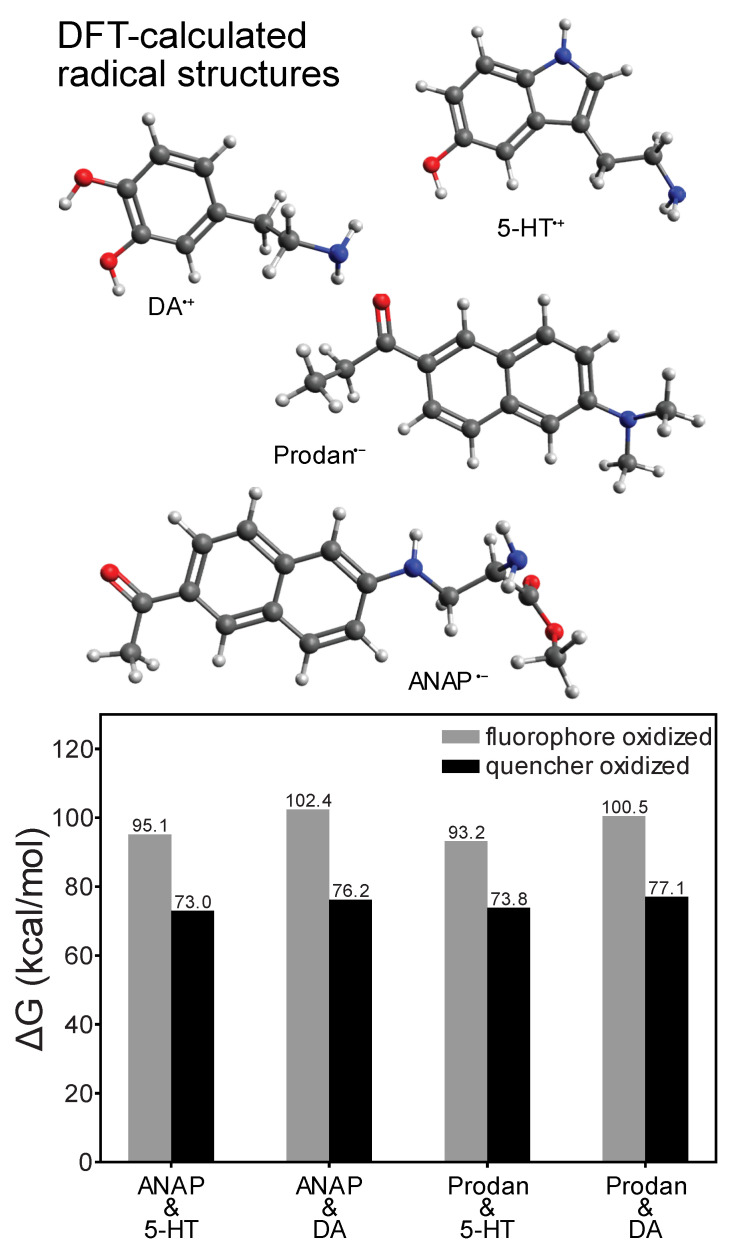
DFT-calculated redox ∆G values for electron transfer reactions of l-ANAP methyl ester or Prodan with dopamine or 5-HT (ωB97XD density functional and def2-TZVPP basis), showing that fluorophore reduction/quencher oxidation is consistently more thermodynamically stable. Structures shown above are those of the most redox-stable radical species.

**Figure 5 ijms-26-10815-f005:**
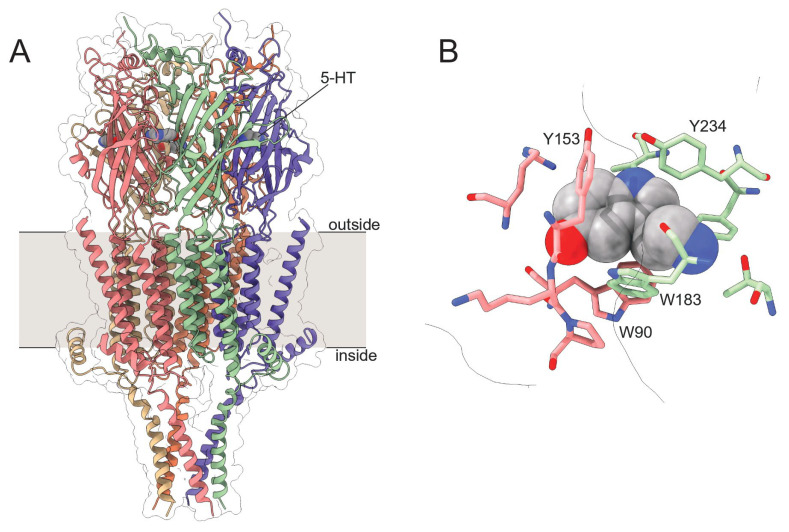
Structure of 5-HT_3A_. (**A**). Cryo-electron microscopy structure of 5-HT_3A_. Subunits are individually colored. 5-HT is shown as a gray, space-filling model. (**B**). One (of five) 5-HT binding sites in 5-HT_3A_ highlighting nearby aromatic residues. Structural models are from Basak et al. (PDB accession number 6DG8) [25].

**Figure 6 ijms-26-10815-f006:**
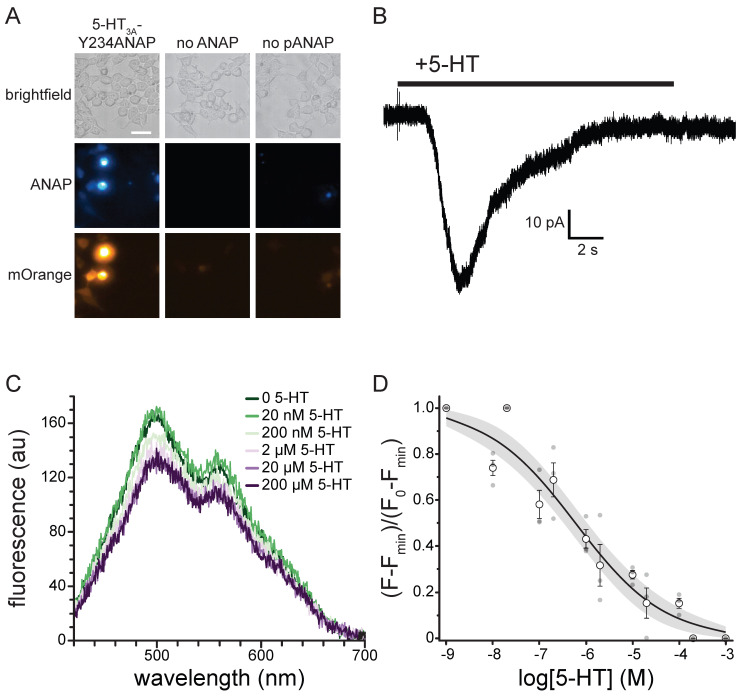
5-HT binding to 5-HT_3A_-Y234ANAP. (**A**). Brightfield and fluorescence micrographs showing cells co-expressing the 5-HT_3A_-Y234TAG plasmid and pANAP and cultured in the presence (left) and absence (middle) of ANAP. The right panel shows cells expressing the 5-HT_3A_-Y234TAG plasmid without pANAP and cultured in the presence of ANAP. The scale bar is 20 µm. (**B**). Representative whole-cell currents in response to bath application of 240 µM 5-HT for a cell expressing 5-HT_3A_-Y234ANAP. (**C**). Fluorescence emission spectra acquired from a cell expressing 5-HT_3A_-Y234ANAP in the presence of increasing concentrations of 5-HT. (**D**). Normalized quenching of 5-HT_3A_-Y234ANAP by 5-HT in HEK293T cells. The data were fit to a modified Hill equation (see Materials and Methods) with a slope of −0.46 and EC_50_ value of 594 nM. The shaded area shows the 95% confidence interval for the fit. *n* = 6 cells (3 replicates at each concentration).

## Data Availability

The original contributions presented in this study are included in the article/Appendix A. Further inquiries can be directed to the corresponding author.

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
