# Peer review of "Measuring Serotonin Binding to Its Receptors In Vitro via Charge Transfer to ANAP"

_ijms, 2025, doi:10.3390/ijms262210815_

Round 1
Reviewer 1 Report
Comments and Suggestions for Authors
The manuscript entitled “Measuring serotonin binding to its receptors via charge transfer to ANAP” written by Puljung and coworkers and submited to Internation Journal of Molecular Sciences deals with the study of serotonin (5-HT) binding to 5-HT receptors via a fluorescence-based methodology. The fluorescence quenching of the non-canonical amino acid L-3-(6-acetylnaphthalen-2-ylamino)-2-aminopropanoic acid (ANAP) in the presence of serotonin served as a mechanism to measure serotonin binding to 5-HT receptors in cells. The ANAP quenching process was studied in solution by time resolved photoluminescence and transient absorption spectroscopy, revealing a concentration-dependent process. DFT calculations confirmed an electron transfer mechanism. The process of ANAP quenching by serotonin was used to study 5-HT binding process in ANAP-tagged channels expressed in intact cells. The presented work continues the author’s previous work on labeling proteins with ANAP fluorescent tag (Puljung et al. Activation mechanism of ATP-sensitive K+ channels explored with real-time nucleotide binding eLife 8:e41103). They studied binding of trinitrophenyl-ATP (TNP-ATP) derivatives to ATP-sensitive K+ channels (KATP) in nucleotide-binding domains (NBD). The binding studied relied on FRET efficiency between the ANAP and fluorescent TNP-ATP derivatives.
While the authors present a solid study with complete experimental details, there are some issues that must be clarified before acceptance for publication.
The y term of the Sern-Volmer equation (equation 2) is F0/F and not F/F0. Therefore, in Figure 2C and 2E, the y axis should be F0/F instead of F/F0. The slope should still be positive if fluorescence quenching is observed.
In Figure 1B, it can be observed that low concentrations of 5-HT cause almost no changes on the fluorescence emission. 250 equivalents were added to observe a first change in the fluorescence emission. Could the authors comment on the large excess of 5-HT (up to 2500 eq) needed to quench the fluorescence and compare with other reported systems?
In Figure 1D, it seems that at low concentration of DA a small increase in fluorescence was observed before quenching with higher concentrations of DA. Can the authors comment on this phenomenon? This phenomenon is responsible for the low R0 of the Stern-Volmer linear regression plot.
Time resolved spectra from the SI (Figure S3) should be presented with χ2 and residuals to evaluate the goodness of the fit.
The sentence at line 333-335 is not justified with data “Our TRPL measurements of emission from a mixture of both compounds exhibited the expected individual 5-HT and ANAP signal components but with decreased overall intensity.” How many equivalents of 5-HT were used. Could the authors give more information about the mentioned “decreased overall intensity”? The results of the time resolved quenching experiments are important in order to differentiate static from dynamic quenching, however the fluorescence of 5-HT may be problematic to obtain reliable data. The authors should comment on this.
The abstract mention a rapid reversible charge transfer (line 17) and the discussion section introduces the term “back electron transfer” (line 551). This reported time for the studied process was 3.2 ns. The authors should clarify the manuscript with respect to the terms used. Furthermore, a rapid process would be expected in the ps time frame.
The purpose of comparing ANAP with Prodan is as radical quenchers in DFT studies is not clear. The authors should add a sentence on that.
The authors studied DA agonist, another ANAP quencher in 5-HT receptors, the study performed in solution and DFT calculations. Was it possible to study DA in in ANAP-tagged channels expressed in intact cells.
Minor comments:
The reviewer suggests title should be reformulated
X axes of all Figures with emission spectra (Figure 1C, 1D, 5C, S1, S2A, S2B, S3A andS3C) should be relabeled: “Wavelength (nm)” instead of “nm”
X axes of Figures 2 should be relabeled: “Time Delay (ns)” instead of “ns”
X axes of Figures S4 should be relabeled: “Time (ns)” instead of “ns”
Comments on the Quality of English Language
English language is acceptable; however, some sentences need refinement to facilitate better understanding and optimize manuscript impact. Some examples of sentences to modify are given below:
“Consistent with this, we observed a small, reproducible shift the peak wavelength for ANAP emission from 451 nm to 447 nm in the presence of 100 mM 5-HT (Fig. 1B).
“While FRET or Dexter quenching mechanisms would result in the fluorophore/quencher pair being completely returned to their electrically neutral electronic ground states following the quench, the existence of a longer-lived 3.18 ns signal is consistent with the formation of a charge-transfer state that persists in solution after the quench event has occurred”
Author Response
We thank the reviewers for their attention to detail and helpful comments. In response to these comments, we have made several additions to the manuscript (highlighted in red), performed additional experiments, and added/updated 4 figures (Figures 2, Figure S1B, Figure S1C, and Figure S4). We hope you will find this manuscript greatly improved over our initial submission. Below, please find the responses to your individual comments.
Reviewer #1—we thank Reviewer #1 for their careful attention and helpful suggestions, particularly in regard to our time-resolved spectroscopy. Below, please find our responses.
Comments and Suggestions for Authors
The manuscript entitled “Measuring serotonin binding to its receptors via charge transfer to ANAP” written by Puljung and coworkers and submitted to Internation Journal of Molecular Sciences deals with the study of serotonin (5-HT) binding to 5-HT receptors via a fluorescence-based methodology. The fluorescence quenching of the non-canonical amino acid L-3-(6-acetylnaphthalen-2-ylamino)-2-aminopropanoic acid (ANAP) in the presence of serotonin served as a mechanism to measure serotonin binding to 5-HT receptors in cells. The ANAP quenching process was studied in solution by time resolved photoluminescence and transient absorption spectroscopy, revealing a concentration-dependent process. DFT calculations confirmed an electron transfer mechanism. The process of ANAP quenching by serotonin was used to study 5-HT binding process in ANAP-tagged channels expressed in intact cells. The presented work continues the author’s previous work on labeling proteins with ANAP fluorescent tag (Puljung et al. Activation mechanism of ATP-sensitive K+ channels explored with real-time nucleotide binding eLife 8:e41103). They studied binding of trinitrophenyl-ATP (TNP-ATP) derivatives to ATP-sensitive K+ channels (KATP) in nucleotide-binding domains (NBD). The binding studied relied on FRET efficiency between the ANAP and fluorescent TNP-ATP derivatives.
While the authors present a solid study with complete experimental details, there are some issues that must be clarified before acceptance for publication.
The y term of the Sern-Volmer equation (equation 2) is F0/F and not F/F0. Therefore, in Figure 2C and 2E, the y axis should be F0/F instead of F/F0. The slope should still be positive if fluorescence quenching is observed.
Thank you for noticing this mistake. We have now correctly labeled the axes in Figure 1 and Figure S3. In response to Reviewer #2, we have also added error bars to the Stern-Volmer plots.
In Figure 1B, it can be observed that low concentrations of 5-HT cause almost no changes on the fluorescence emission. 250 equivalents were added to observe a first change in the fluorescence emission. Could the authors comment on the large excess of 5-HT (up to 2500 eq) needed to quench the fluorescence and compare with other reported systems?
The results presented in Figure 1 represent dynamic quenching in solution via electron transfer. As this process requires the donor and quencher to be within contact distance, measurements in solution required very high quencher concentrations. In the context of a receptor, selective binding of 5-HT in the ANAP-labeled binding pocket will bring the fluorophore/quencher pair into close proximity at much lower quencher concentrations. We have made additions to the manuscript on page 16 (starting with line 326) to make this point more clearly. This also includes a discussion of physiological serotonin concentrations.
In Figure 1D, it seems that at low concentration of DA a small increase in fluorescence was observed before quenching with higher concentrations of DA. Can the authors comment on this phenomenon? This phenomenon is responsible for the low R0 of the Stern-Volmer linear regression plot.
We attribute this effect to the environmental sensitivity of ANAP and observe a similar increase in fluorescence at low concentrations of 5-HT. We have added additional text pointing out this effect as it pertains to DA (page 7, starting on line 133), and have included a discussion of this effect in relation to our measurements in ANAP-labeled receptors (page 17, line 366). The increase in fluorescence at low quencher concentrations was comparable for 5-HT and DA. Therefore, we conclude that the shallower slope for our Stern-Volmer plot reflects a bona fide difference in the quenching efficiency between 5-HT and DA.
Time resolved spectra from the SI (Figure S3) should be presented with χ2 and residuals to evaluate the goodness of the fit.
We have added residuals and χ2 values to Figure S4 to demonstrate the quality of our fits; we have further addressed fit quality and the trade-offs associated with our relatively low-parameter model with an added section in the text near the beginning of the time-resolved spectroscopy section (page 8, line 145).
The sentence at line 333-335 is not justified with data “Our TRPL measurements of emission from a mixture of both compounds exhibited the expected individual 5-HT and ANAP signal components but with decreased overall intensity.” How many equivalents of 5-HT were used. Could the authors give more information about the mentioned “decreased overall intensity”? The results of the time resolved quenching experiments are important in order to differentiate static from dynamic quenching, however the fluorescence of 5-HT may be problematic to obtain reliable data. The authors should comment on this.
We have expanded this portion of the manuscript to address the time-resolved ANAP quenching concerns, including specifying the concentrations used and adding a new figure (S4) that compares quenched versus unquenched ANAP. We appreciate the reviewer’s suggestion of addressing static versus dynamic quenching and have clarified our analysis of the quenching mechanism (page 8, line 159 ff.).
The abstract mention a rapid reversible charge transfer (line 17) and the discussion section introduces the term “back electron transfer” (line 551). This reported time for the studied process was 3.2 ns. The authors should clarify the manuscript with respect to the terms used. Furthermore, a rapid process would be expected in the ps time frame.
The “rapid reversible charge transfer” description from line 17 was meant to imply rapid on the timescale of biological processes (e.g. dissociation of transmitter from a receptor) that might be studied via the use of ANAP, rather than rapid on the timescale of electron transfer. We have updated the language in the abstract to clarify the timescale and mechanism and introduce the term back-electron transfer earlier in both the abstract and introduction. While back-electron transfer is a broadly studied phenomenon, we have added an additional citation to the Discussion section regarding back-electron transfer should readers be interested in further background on the topic.
The purpose of comparing ANAP with Prodan is as radical quenchers in DFT studies is not clear. The authors should add a sentence on that.
We included Prodan in our calculations as ANAP’s side chain is structurally similar to Prodan. We have included an additional sentence on page 11 (line 213) to clarify.
The authors studied DA agonist, another ANAP quencher in 5-HT receptors, the study performed in solution and DFT calculations. Was it possible to study DA in in ANAP-tagged channels expressed in intact cells.
Yes, we believe this is possible. We are excited to explore this in a future manuscript.
Minor comments:
The reviewer suggests title should be reformulated
In response to this comment and one of the other reviewers, we have updated the title and hope you find it suitable.
X axes of all Figures with emission spectra (Figure 1C, 1D, 5C, S1, S2A, S2B, S3A andS3C) should be relabeled: “Wavelength (nm)” instead of “nm”
We made this change.
X axes of Figures 2 should be relabeled: “Time Delay (ns)” instead of “ns”
We made this change.
X axes of Figures S4 should be relabeled: “Time (ns)” instead of “ns”
We made this change.
Comments on the Quality of English Language
English language is acceptable; however, some sentences need refinement to facilitate better understanding and optimize manuscript impact. Some examples of sentences to modify are given below:
We have made edits to the sentences in question and have edited the entire manuscript to maximize clarity.

Reviewer 2 Report
Comments and Suggestions for Authors
Puljung and coworkers reported ANAP can be used to detect 5-HT or DA binding to its receptors. The topic is interesting and important. The fluorescence of ANAP was quenched in solution by both 5-HT and DA, which involved a rapidly reversible charge-transfer process. In addition, addition of 5-HT caused a concentration-dependent quenching of fluorescence from ANAP-tagged channels expressed in intact cells and unroofed plasma membranes, demonstrating the utility of this method for measuring 5-HT binding to its receptors. The manuscript is well organized and supported by the data. However, several issues should be claimed before publication.
- Although the fluorescence of ANAP can be quenched by 5-HT, it can also be quenched by DA. Moreover, the quenching effects show no significant difference—only variations in concentration. This constitutes a critical flaw for ANAP as a probe for the selective detection of 5-HT. Therefore, I believe the theme of this article should be appropriately revised.
- For Figure 1 and Figure S1, quenching of ANAP by 5-HT and DA should be test in buffered solution, not in DMSO solution or water.
- The author's study demonstrates that even using a high concentration of 5-HT (far exceeding the normal levels of 5-HT in humans), it cannot fully quench the fluorescence of ANAP, indicating poor sensitivity of ANAP as probe. Furthermore, certain diseases can reduce 5-HT levels in the body—under such conditions, whether this concentration would still quench the fluorescence of probe remains unclear.
- In Figures 1C, 1E, 5D, S3B, S3D and S5 should be added the error bars.
- There are some typing or grammar errors. They should be corrected thoroughly. For example, in the caption of Figure 1, the numbers in "M-1" and "R2" should be superscripted.
Author Response
We thank the reviewers for their attention to detail and helpful comments. In response to these comments, we have made several additions to the manuscript (highlighted in red), performed additional experiments, and added/updated 4 figures (Figures 2, Figure S1B, Figure S1C, and Figure S4). We hope you will find this manuscript greatly improved over our initial submission. Below, please find the responses to your individual comments.
Reviewer #2—thank you for your comments. In response, we have included new data and analysis and made major revisions to the text. Below, you will find our responses to your specific comments in bold.
Comments and Suggestions for Authors
Puljung and coworkers reported ANAP can be used to detect 5-HT or DA binding to its receptors. The topic is interesting and important. The fluorescence of ANAP was quenched in solution by both 5-HT and DA, which involved a rapidly reversible charge-transfer process. In addition, addition of 5-HT caused a concentration-dependent quenching of fluorescence from ANAP-tagged channels expressed in intact cells and unroofed plasma membranes, demonstrating the utility of this method for measuring 5-HT binding to its receptors. The manuscript is well organized and supported by the data. However, several issues should be claimed before publication.
- Although the fluorescence of ANAP can be quenched by 5-HT, it can also be quenched by DA. Moreover, the quenching effects show no significant difference—only variations in concentration. This constitutes a critical flaw for ANAP as a probe for the selective detection of 5-HT. Therefore, I believe the theme of this article should be appropriately revised.
Thank you for your comments. Our method is intended for in vitro measurements of receptor occupancy/binding affinity and not for use in whole organisms. In such studies, we are less concerned about the overlap in 5-HT and DA effects, as one can measure the binding of 5-HT and DA in separate experiments.
In response to your comment we have
- Modified the title of the manuscript
- Included text throughout the manuscript to clarify this point.
- Included a discussion on in vivo 5-HT detection (page 19) and physiological 5-HT concentrations (page 16).
- For Figure 1 and Figure S1, quenching of ANAP by 5-HT and DA should be test in buffered solution, not in DMSO solution or water.
Thank you for your comment. We have expanded Figure S1 to show quenching of ANAP by 5-HT in both phosphate buffered saline and our recording buffer. We added additional text on page 6 of the Results section and amended the figure legends and methods accordingly.
- The author's study demonstrates that even using a high concentration of 5-HT (far exceeding the normal levels of 5-HT in humans), it cannot fully quench the fluorescence of ANAP, indicating poor sensitivity of ANAP as probe. Furthermore, certain diseases can reduce 5-HT levels in the body—under such conditions, whether this concentration would still quench the fluorescence of probe remains unclear.
We have added discussion of this incomplete quenching on page 17 (starting on line 360). Despite the incomplete quenching, our method provides a sufficiently large signal to provide repeatable measurements in in vitro studies in cells and unroofed cell membranes.
- In Figures 1C, 1E, 5D, S3B, S3D and S5 should be added the error bars.
We have added these error bars.
- There are some typing or grammar errors. They should be corrected thoroughly. For example, in the caption of Figure 1, the numbers in "M-1" and "R2" should be superscripted.
Thank you. We have made this change and proofread the manuscript for errors.

Reviewer 3 Report
Comments and Suggestions for Authors
In this work, the authors proposed a procedure of measuring serotonin binding to its receptors via quenching the fluorescence of associated protein (ANAP). The manuscript is consistent and easy to read but there are serious issues causing a need of major revision. These issues contain in the comments below.
1) In the Introduction, a discussion of physiological concentrations of 5-HT is missed. Detection limit and accuracy required for practical analysis must be indicated.
2) More comparison with existing techniques of 5-HT measuring should be included also.
3) One of the main questions to whole manuscript is the used concentration range of 5-HT which is outside of physiological one. Data presented indicate that the proposed method may not work at lower concentrations and then its applicability is lost.
4) Emission spectra of ANAP acquired in physiologically related media (PBS buffer, nutrient medium) have to be presented.
5) The authors have to verify the accuracy of lifetime measurements (0.001 ns). Such accuracy contradicts particularly with the statement that the lifetime of 5-HT (2.231 ± 0.001 ns) is broadly consistent with published value for 5-HT (4 ns). Overall, the results of time-resolved photoluminescence are unclear since there are no good match with literature and benefits in identifying the interactions between ANAP and 5-HT.
6) The DFT calculations did not show any significant difference between 5-HT and DA while K(SV) for them differ by almost 2 times. How the authors can explain this?
7) What is the measurement inaccuracy of intensity in the emission spectra given in the part C of Figure 5? The working range of 20 nm – 200 mkM (4 orders) corresponds to only 25% drop in fluorescence intensity. These data indicate the extremely low accuracy of the proposed method.
8) In in vitro study, negative controls are not provided. A specificity of the response obtained has not been demonstrated either.
Minor comments:
- The scale marker in the fluorescence images is not labeled in Figure 5.
- Line 454 in Page 14 contains a typo, it should be 240 mkM obviously.
Author Response
We thank the reviewers for their attention to detail and helpful comments. In response to these comments, we have made several additions to the manuscript (highlighted in red), performed additional experiments, and added/updated 4 figures (Figures 2, Figure S1B, Figure S1C, and Figure S4). We hope you will find this manuscript greatly improved over our initial submission. Below, please find the responses to your individual comments.
Reviewer #3—thank you for your helpful comments. We have seriously considered your points and included new figures and major additions to the text. Our responses to your specific comments are in bold below.
Comments and Suggestions for Authors
In this work, the authors proposed a procedure of measuring serotonin binding to its receptors via quenching the fluorescence of associated protein (ANAP). The manuscript is consistent and easy to read but there are serious issues causing a need of major revision. These issues contain in the comments below.
- In the Introduction, a discussion of physiological concentrations of 5-HT is missed. Detection limit and accuracy required for practical analysis must be indicated.
Thank you for this comment. We have added a discussion on physiological concentrations of 5-HT in the gut, blood, brain, and the synapse and the relevance of our binding assay to these concentrations (page 16). The concentration range over which we measured binding is consistent with the range reported for 5-HT activation of 5-HT3 receptors.
We have also included additional wording and amended our title to stress that our assay is intended for in vitro studies of 5-HT binding to its receptors in heterologous expression systems, and not as a low-threshold 5-HT detector.
- More comparison with existing techniques of 5-HT measuring should be included also.
We have now included a discussion of existing in vitro assays to measure 5-HT binding to its receptors beginning on page 18.
- One of the main questions to whole manuscript is the used concentration range of 5-HT which is outside of physiological one. Data presented indicate that the proposed method may not work at lower concentrations and then its applicability is lost.
Thank you for your comment. Reviewer #1 shared this concern. Below is our response to that reviewer.
The results presented in Figure 1 represent dynamic quenching in solution via electron transfer. As this process requires the donor and quencher to be within contact distance, measurements in solution required very high quencher concentrations. In the context of a receptor, selective binding of 5-HT in the ANAP-labeled binding pocket will bring the fluorophore/quencher pair into close proximity at much lower quencher concentrations. We have made additions to the manuscript on page 16 (starting with line 326) to make this point more clearly. This also includes a discussion of physiological serotonin concentrations.
- Emission spectra of ANAP acquired in physiologically related media (PBS buffer, nutrient medium) have to be presented.
We have now verified that 5-HT quenches ANAP in PBS solution as well as in our recording buffer. These data are now included in Figure S1, with additional text on page 6 of the Results section and amended the figure legends and methods accordingly.
5) The authors have to verify the accuracy of lifetime measurements (0.001 ns). Such accuracy contradicts particularly with the statement that the lifetime of 5-HT (2.231 ± 0.001 ns) is broadly consistent with published value for 5-HT (4 ns). Overall, the results of time-resolved photoluminescence are unclear since there are no good match with literature and benefits in identifying the interactions between ANAP and 5-HT.
The section regarding TRPL lifetimes has been rewritten to clarify these issues (page 8). Fluorophore lifetimes are expected to vary in different solvents, particularly due to the effect of solvent dielectric as well as pH-dependent effects. We have removed reference to “broadly consistent” lifetimes to avoid confusion. We have also clarified that our accuracies are characterizing only the fit of the convolved model to the data, rather than overall uncertainty; our actual lifetime uncertainty is most likely limited primarily by the 0.4 ns time bins of our instrument. We believe the improved analysis of dynamic quenching addresses the final portion of this comment, showing clearer benefits to the application of TRPL for understanding ANAP/5-HT interactions.
6) The DFT calculations did not show any significant difference between 5-HT and DA while K(SV) for them differ by almost 2 times. How the authors can explain this?
Our DFT calculations describe the relative stabilities of ground-state species before and after the electron transfer process has occurred; these calculations provide no direct modeling of the interaction of excited-state ANAP and 5-HT or DA at the moment quenching occurs. Such calculations would require time-dependent DFT (TD-DFT) calculations that are computationally expensive and beyond the scope of this work. Nonetheless, our calculations do demonstrate that the radical ion pairs are less stable than the neutral ground state species, explaining the source of the driving force behind the reversibility of the charge transfer. These calculations further show that light sources used for this experiment do not provide enough energy to produce certain radical ion pairs, providing support for the proposed directionality of the charge transfer. We have added language to the DFT section (page 10) to clarify the scope of conclusions our calculations can provide.
7) What is the measurement inaccuracy of intensity in the emission spectra given in the part C of Figure 5? The working range of 20 nm – 200 mkM (4 orders) corresponds to only 25% drop in fluorescence intensity. These data indicate the extremely low accuracy of the proposed method.
We believe that our inability to achieve 100% quenching in ANAP-tagged channels reflects the short distance dependence between of quenching, and/or an unfavorable orientation between ANAP in the 5-HT3A receptor and 5-HT in the binding site. Whereas quenching was incomplete and variable, we were able to obtain repeatable measurements in intact cells as well as in unroofed membranes. We have now included a discussion of these limitations on page 17.
8) In in vitro study, negative controls are not provided. A specificity of the response obtained has not been demonstrated either.
We share this concern. We have amended the manuscript to clarify that this assay is intended for in vitro measurements of 5-HT binding to its receptors. In our in vitro studies, we are confident in the specificity of our measurements for two reasons:
- As we only performed measurements for cells/membranes that exhibited both ANAP and mOrange fluorescen, we are confident that the signal we measured came from ANAP labeled receptors.
- As we have established that ANAP is quenched via a short-range electron transfer mechanism, 5-HT in solution or bound elsewhere in the cell would not affect the ANAP fluorescence.
We have added an extensive discussion of the advantages and limitations of our method to the text..
Minor comments:
- The scale marker in the fluorescence images is not labeled in Figure 5.
Thank you for noticing. We have added the scale to the figure legend.
- Line 454 in Page 14 contains a typo, it should be 240 mkM obviously.
The current trace in the figure is in 240 µM, but our average results are reported at a concentration of 24 µM 5-HT.

Round 2
Reviewer 2 Report
Comments and Suggestions for Authors
The authors have addressed most of the issues raised by the reviewers and the manuscript is much improved. The revised manuscript is suitable for publication.
Reviewer 3 Report
Comments and Suggestions for Authors
The authors have addressed most of reviewing comments, and the paper quality has been improved. The work is suitable for accepted in the journal now.